# Superior zero thermal expansion dual-phase alloy via boron-migration mediated solid-state reaction

Chengyi Yu[1], Kun Lin [1] ✉, Xin Chen [1], Suihe Jiang [1], Yili Cao[1], Wenjie Li[1], Liang Chen[1], Ke An [2], Yan Chen[2], Dunji Yu[2], Kenichi Kato[3], Qinghua Zhang[4], Lin Gu[4], Li You[1], Xiaojun Kuang [5], Hui Wu [6], Qiang Li[1], Jinxia Deng[1] & Xianran Xing [1] ✉

Rapid progress in modern technologies demands zero thermal expansion (ZTE) materials with multi-property profiles to withstand harsh service conditions. Thus far, the majority of documented ZTE materials have shortcomings in different aspects that limit their practical utilization. Here, we report on a superior isotropic ZTE alloy with collective properties regarding wide operating temperature windows, high strength-stiffness, and cyclic thermal stability. A boron-migration-mediated solid-state reaction (BMSR) constructs a salient "plum pudding" structure in a dual-phase Er-Fe-B alloy, where the precursor $ErFe_{10}$ phase reacts with the migrated boron and transforms into the target $Er_2Fe_{14}B$ (pudding) and $\alpha$-Fe phases (plum). The formation of such microstructure helps to eliminate apparent crystallographic texture, tailor and form isotropic ZTE, and simultaneously enhance the strength and toughness of the alloy. These findings suggest a promising design paradigm for comprehensive performance ZTE alloys.

Zero thermal expansion (ZTE) is indispensable in high-precision technological applications, ranging from optical components to aerospace structures[1-5]. Intermetallic compounds, the main branch of ZTE materials, have garnered increasing interest for their distinctive metallicity[6-12]. While possessing ZTE property solely is not enough for practical applications[13-19]. A desirable ZTE metal material should be a multi-property profile: (i) a wide ZTE temperature window ($\Delta T$, covering room temperature), which can resist dimensional changes caused by large temperature fluctuations[20,21]; (ii) sufficient mechanical response (strength, stiffness, ductility, etc.) to withstand required mechanical loads[22-24]; (iii) isotropic ZTE performance (three-dimensional size stability). The anisotropic thermal expansion property will restrict the range of the material application[25,26]; (iv) cyclic thermal stability, i.e., the structure and thermal expansion characteristics are stable in the process of resisting thermal shock, etc. Unfortunately, so far few materials could satisfy these requirements simultaneously.

Since the thermal expansion is often coupled to magnetic interaction in metallic alloys, it is feasible to manipulate the coefficient of thermal expansion through chemical modifications. Owing to the drastic magnetic or structural phase transition, ZTE occurs predominantly at low temperatures (lower than 300 K) and its temperature windows are usually narrow, such as in $La(Fe, Si)_{13}$-type[11,27], $Ni_2In$-type[10,28,29] and $(Hf, Ta)Fe_2$ series compounds[7,30]. In $ReCo_2$[31-33] and $Re_2Fe_{17}$ (Re, rare earth element) series compounds[20,21], broad

[1]Beijing Advanced Innovation Center for Materials Genome Engineering, and Institute of Solid State Chemistry, University of Science and Technology Beijing, Beijing 100083, China. [2]Neutron Scattering Division, Oak Ridge National Laboratory, Oak Ridge, TN, USA. [3]RIKEN SPring-8 Center, 1-1-1 Kouto, Sayo-Cho, Sayo-gun, Hyogo 679-5148, Japan. [4]Beijing National Laboratory for Condensed Matter and Institute of Physics, Chinese Academy of Sciences, Beijing 100190, China. [5]Guangxi Key Laboratory of Electrochemical and Magnetochemical Functional Materials, College of Chemistry and Bioengineering, Guilin University of Technology, Guilin 541004, P. R. China. [6]NIST Center for Neutron Research, National Institute of Standards and Technology, Gaithersburg, MD 20899-6102, US. ✉e-mail: kunlin@ustb.edu.cn; xing@ustb.edu.cn

temperature windows of ZTE were achieved. Frustratingly, these ZTE materials are inherently brittle and of low strength (<200 MPa), resulting in few practical uses. Although artificial composites can improve their mechanical properties[34,35], the material fatigue fails in the process of resisting thermal shock and will cause catastrophic damage due to the weak interfacial bonding within the composites[36,37]. Recently, a series of dual-phase ZTE alloys with appropriate mechanical characteristics have been designed and prepared through eutectic reactions[38,39], such as Ho-Fe[13], Er-Fe-V-Mo[40], La-Fe-Si[41,42], etc. Alternatively, the microstructure of NiTi alloys[25,43], and Mn$_{5-x}$Fe$_x$Si$_3$ compounds[44], is manipulated to tailor their two-dimensional (2D) ZTE performance. While the overall ZTE alloys are still suboptimal as they exhibit either strong anisotropic thermal expansion or a narrow ZTE temperature window. It is still a daunting challenge to design and develop a multi-property profile stable ZTE alloy[45–48].

In this work, we aim to face the dilemma head-on. By incorporating a boron-migration-mediated solid-state reaction (ErFe$_{10}$ + B$_{GBs}$ → Er$_2$Fe$_{14}$B + α-Fe; grain boundaries, GBs) into an Er-Fe-B ternary alloy, we were able to architect an intriguing "plum pudding" microstructure. It was composed of an Er$_2$Fe$_{14}$B matrix (E phase, pudding)

with a negative thermal expansion (NTE) and two distinct types of α-Fe (α phase, plum) with a positive thermal expansion (PTE): the primary α phase at the grain boundaries (GBs) and the dispersed intragranular α phase in submicron size. The "plum pudding" microstructure enables the brittle E phase to be toughened at both GBs and intragranular grains, and its mechanical properties are remarkably improved over the pure E phase[37,49,50]. More importantly, the solid-state reaction removes the crystallographic texture of the parent phase, resulting in the formation of the three-dimensional isotropic ZTE. The present strategy sheds light on designing and synthesizing metallic materials for structural applications[51].

## Results and Discussion
### Microstructure and crystal structure
Figure 1a illustrates the overall boron-migration-mediated solid-state reaction (BMSR): ErFe$_{10}$ + B$_{GBs}$ (precursor alloy) → Er$_2$Fe$_{14}$B + α-Fe (target alloy). First, a precursor alloy (ErFe$_{10}$ + B$_{GBs}$, labeled as Pre. Er-Fe-B, GBs: grain boundaries) was generated by annealing the as-cast sample at 1473 K (1 day) as shown in Fig. 1b and Supplementary Fig. 1. It is composed of the primary ErFe$_{10}$ (~200 μm) phase and the

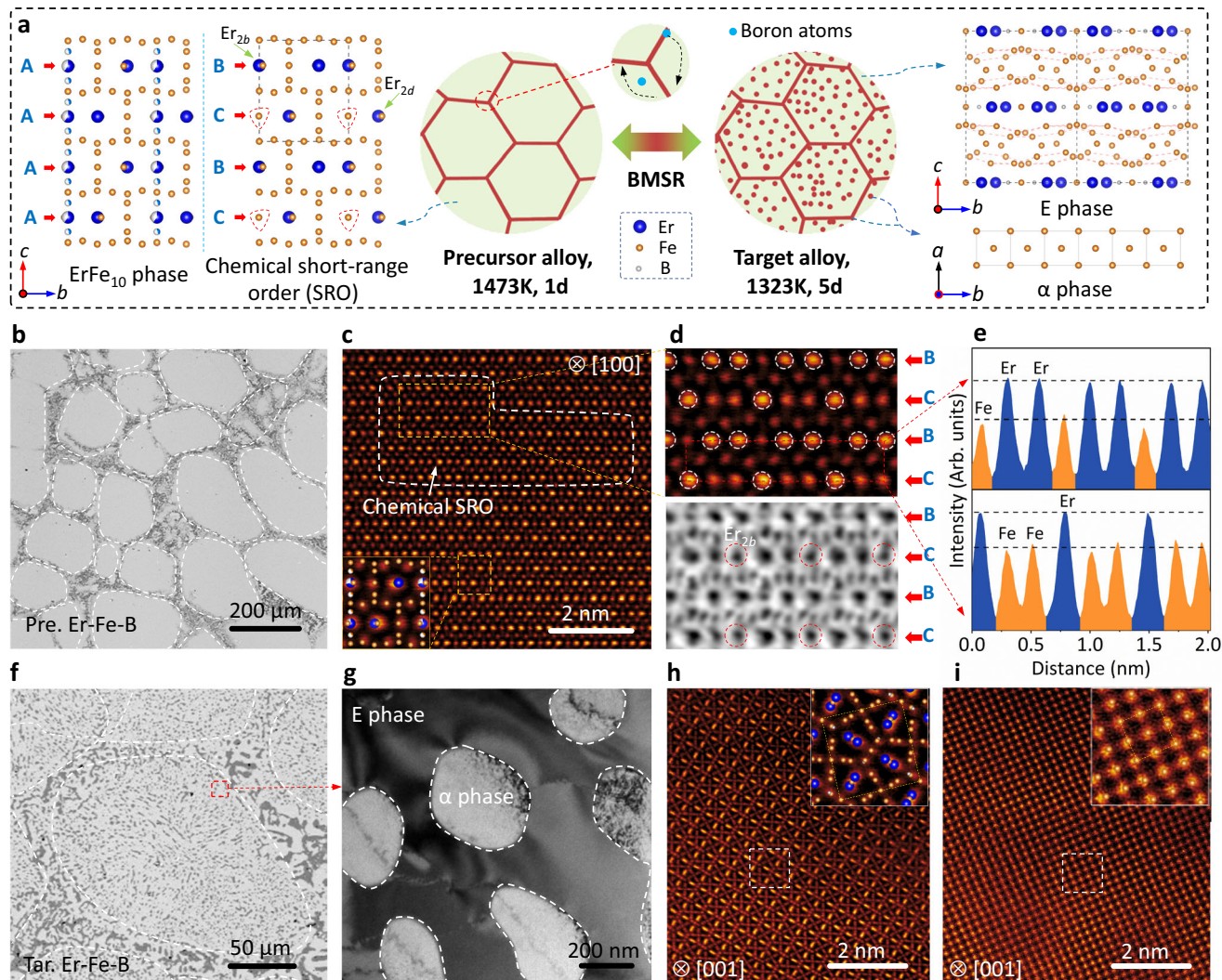

**Fig. 1 | The microstructure of the ZTE alloy. a** Mechanism of boron-migration-mediated solid–state reaction (BMSR) and the crystal structure model of the phases. The structure models of ErFe$_{10}$ phase, E phase and α phase. **b** The microstructure of the Pre. Er-Fe-B is determined by an electro-probe microanalyzer (EPMA). **c** The HAADF-STEM image of the ErFe$_{10}$ phase along the [100] zone axis, the inset is an enlarged view of the disordered atomic structure. The white

rectangle is the area of chemical short-range order (SRO). **d** The enlarged HAADF-STEM and annular bright-field (ABF) images of the SRO area marked in (**c**). **e** The intensity profile of B-C layers in (**d**). **f** The microstructure of the Tar. Er-Fe-B determined by EPMA. **g** TEM images of the Tar. Er-Fe-B. **h-i** The HAADF-STEM image of the E phase and α phase along [001] zone axes.

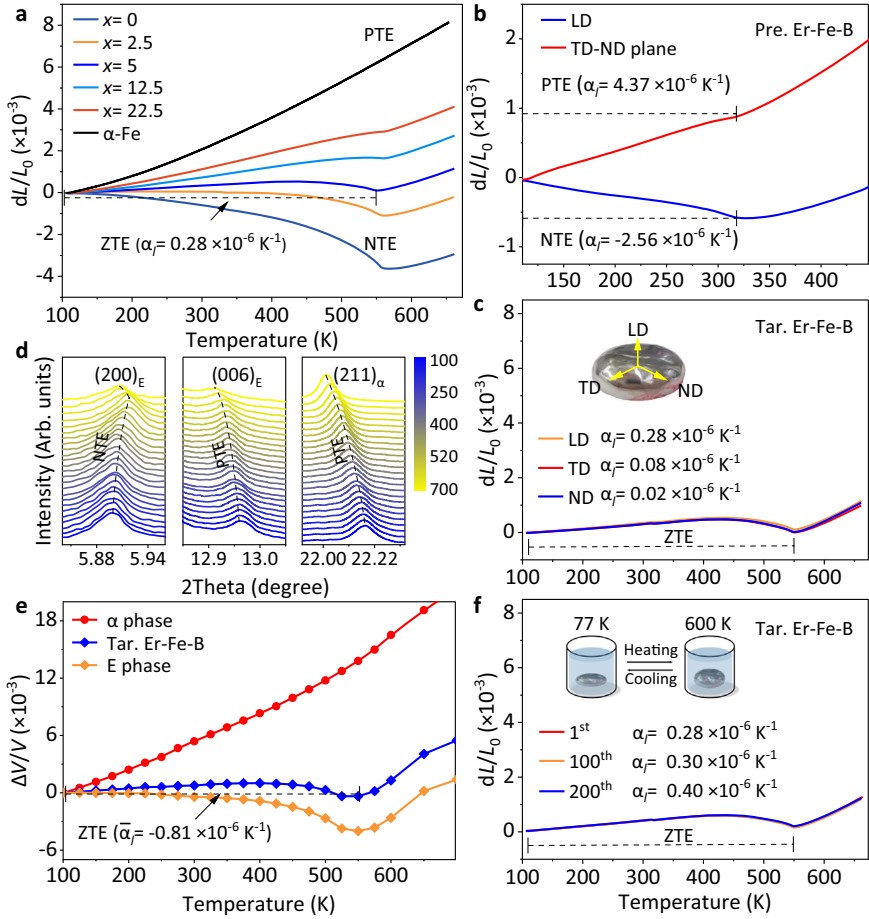

**Fig. 2 | Thermal expansion performance of the series alloys. a** The dilatometer thermal expansion property of $Er_2Fe_{14+x}B_{1+0.07x}$ alloys and pure iron. **b** The dilatometer thermal expansion of the Pre. Er-Fe-B (ZTE composition). **c** The dilatometer thermal expansion of the Tar. Er-Fe-B alloy along loading direction (LD), transverse direction (TD), and normal direction (ND), respectively. **d** The temperature dependence of $(200)_E$, $(006)_E$, and $(211)_\alpha$ reflections determined by SXRD. **e** The lattice thermal expansion of the Tar. Er-Fe-B alloy. **f** The dilatometer thermal expansions of the Tar. Er-Fe-B alloy in the 1st, 100th, and 200th cycles. Insert the cyclic thermal shock experiment that undergoes a thermal shock from 77 K to 600 K.

intergranular α-Fe and $Fe_2B$ phases (Supplementary Fig. 2 and Supplementary Table 1). The metastable $ErFe_{10}$ is a new phase discovered here, which possesses a 2:17-type topological structure ($P6_3/mmc$) with seven Wyckoff sites ($Er_{2b}$, $Er_{2d}$, $Fe_{4f}$, $Fe_{6g}$, $Fe_{12j}$, $Fe_{12k}$, and $Fe_{4e}$; Supplementary Fig. 3 and Supplementary Table 2). These excess Fe atoms (~16.7 %) mainly replace the $Er_{2b}$ sites disorderly in the manner of Fe-Fe pairs to form $Fe_{4e}$ lattice sites (A types, Fe-Fe pairs, and Er atoms disordered substitution, Fig. 1a), as confirmed by synchrotron X-ray diffraction (Supplementary Fig. 4). Further high-angle annular dark-field (HAADF) and annular bright-field (ABF) images evidence the long-range atomic chemical disorder $Er_{2b}$ site along the <100> zone axis (Fig. 1c and Supplementary Fig. 5). Interestingly, a chemical short-range order (SRO) is observed, where the single Fe atom replaces the $Er_{2b}$ site hierarchically to form B-C-B-C (C: $Er_{2b}$ replaced by a single Fe atom; B: $Er_{2b}$ not replaced, Fig. 1a) layers along the c axis as shown in Fig. 1d, e. Besides, the $Er_{2d}$ site is also partially occupied by the single Fe atoms according to the single crystal diffraction (SCD, Supplementary Fig. 6). We enumerated the structural model of the new $ErFe_{10}$ phase in Supplementary Fig. 7. Such structural singularity is related to the formation of the "plum pudding" microstructure and will be discussed in detail later.

The Pre. Er-Fe-B alloy is subsequently annealed at 1323 K (5 days). The BMSR reaction is proceeded by the selective diffusion of boron atoms: the boron atoms migrate from grain boundaries (GBs) to the interior of the grains of the precursor $ErFe_{10}$ phase upon thermal treatments (Supplementary Fig. 8). The 2:17-type $ErFe_{10}$ matrix

transforms to a 2:14:1-type $Er_2Fe_{14}B$ phase and an intergranular α phase (Supplementary Figs. 9 and 10). The re-precipitated α phase is homogeneously dispersed into the E phase matrix with grain sizes of about 0.1 - 10 μm (Fig. 1f and Supplementary Fig. 11). A "plum pudding" microstructure is architected as a result of such a process (Fig. 1g, plum: α-Fe, labeled as α phase, pudding: $Er_2Fe_{14}B$, labeled as E phase). Further transmission electron microscopy (TEM) observations revealed that parts of the reprecipitated α phase are composed of tiny polycrystalline iron grains (~200 nm, Supplementary Fig. 11 and Fig. 12). The atomic arrangements along the <001> zone axis of the α and E phases clearly exhibit the tetragonal crystal symmetry (space group: $P4_2/mnm$) and cubic crystal symmetry (space group: $I\,m\overline{3}m$), consistent with the diffraction results (Fig. 1h and i).

## Thermal expansion behaviors
The matrix of the E phase exhibits a negative thermal expansion (NTE, Supplementary Fig. 13), whereas the α phase shows a positive thermal expansion[52,53]. Consequently, the total thermal expansion performance of the Er-Fe-B alloy can be facilely tuned by the phase content proportions. Figure 2a is the dilatometer thermal expansion of a series of $Er_2Fe_{14+x}B_{1+0.07x}$ alloys (x = 0.0, 2.5, 5.0 (Tar. Er-Fe-B in the present BMSR), 12.5, and 22.5) with decreasing E/α ratio. It can be tailored from strong NTE of pure E phase ($\alpha_l = -7.95 \times 10^{-6}$ $K^{-1}$ at x = 0.0, 100 − 550 K) to PTE ($\alpha_l = 6.62 \times 10^{-6}$ $K^{-1}$ at x = 22.5, 100 − 550 K). Especially, a remarkable low thermal expansion performance of the Tar. Er-Fe-B (LTE, $\alpha_l = 1.40 \times 10^{-6}$ $K^{-1}$ at x = 5, 100 − 450 K) is attained and it shows

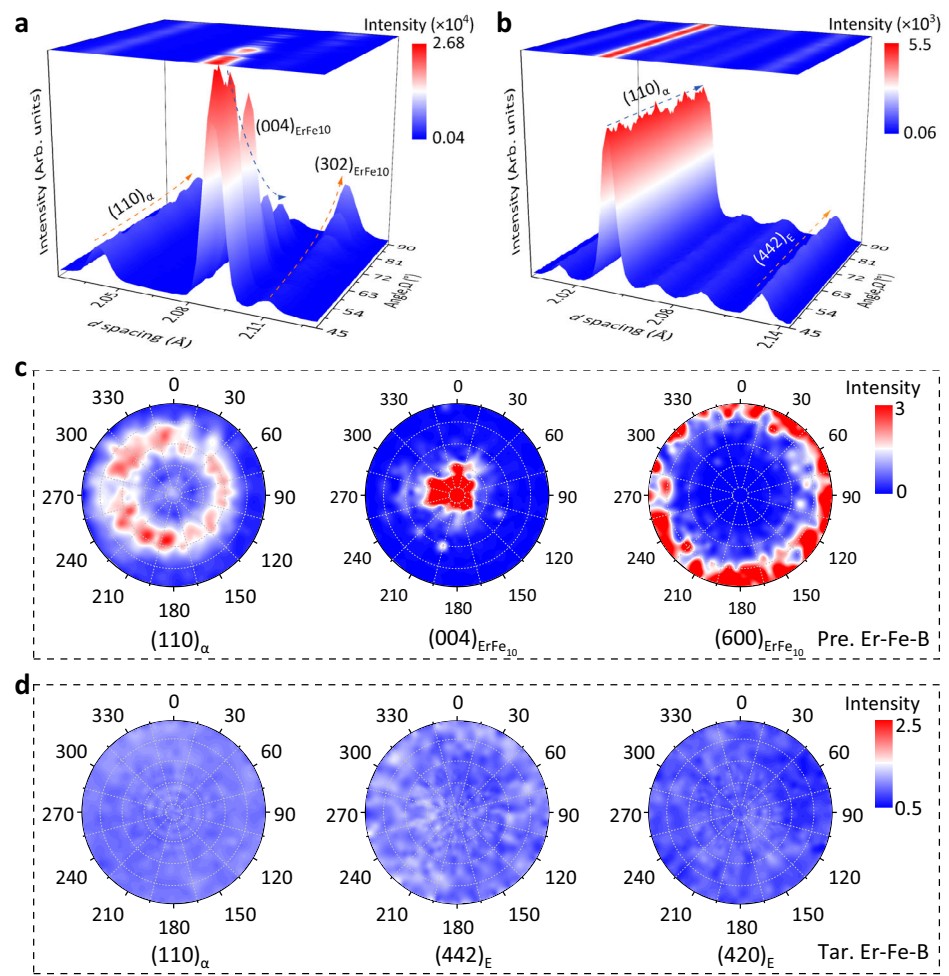

**Fig. 3 | Three-dimensional (3D) crystallographic orientation determined by in-situ neutron diffraction. a, b** The intensity changes of the representative reflections corresponding to the rotation (Ω) of the sample collected in bank 1 detector. **c, d** Pole figures of the representative reflections in the Pre. Er-Fe-B and the Tar. Er-Fe-B.

zero thermal expansion in the temperature window of 100 − 550 K (ZTE, $\alpha_l = 0.28 \times 10^{-6}$ K$^{-1}$). More importantly, in contrast to the Pre. Er-Fe-B displays a strong anisotropy in the thermal expansion (Fig. 2b): NTE along the LD ($\alpha_l = -2.56 \times 10^{-6}$ K$^{-1}$, 100 − 315 K; LD, loading direction) and PTE in the TD · ND plane ($\alpha_l = 4.37 \times 10^{-6}$ K$^{-1}$, 100 − 315 K; transverse direction, TD; normal direction, ND), the dilatometer thermal expansion of the Tar. Er-Fe-B is nearly isotropic in three dimensions (Fig. 2c). Such isotropy of this dual-phase alloy has the potential to greatly expand the area of its applications.

Figure 2d illustrates the evolution of the $(200)_E$, $(006)_E$, and $(211)_\alpha$ synchrotron diffraction peaks of the Tar. Er-Fe-B during heating. The shifting of the $(200)_E$, $(006)_E$, and $(211)_\alpha$ peak positions reveals the lattice contraction along the $a$-axis (NTE) of the E phase and lattice expansion (PTE) of the $\alpha$ phase. The lattice thermal expansions derived from the temperature-dependent synchrotron X-ray diffraction measurements are (Fig. 2e): $\alpha_V = 30.67 \times 10^{-6}$ K$^{-1}$ for the $\alpha$ phase and $\alpha_V = -8.93 \times 10^{-6}$ K$^{-1}$ for the E phase (Supplementary Fig. 14). The apparent lattice thermal expansion of the Tar. Er-Fe-B alloy is $\bar{\alpha}_l \approx \frac{1}{3} \alpha_V = -0.81 \times 10^{-6}$ K$^{-1}$ (100 − 550 K, the specific calculation is provided in Methods), which corroborates to dilatometer measurement. Additional Rietveld refinements quantifying the linear relationship between the content of $\alpha$ phase (mass %) and the coefficient of thermal expansion (Supplementary Figs. 15 and 16). The magnetic measurement confirms that the NTE behavior of both the ErFe$_{10}$ phase and the pure E phase originates from the ferrimagnetic order (Supplementary Fig. 17)[52,53]. Besides, the cyclic thermal shock experiments of the Tar. Er-Fe-B

demonstrates its thermal stability, as the dilatometer thermal expansion remained constant (Fig. 2f) and the microstructure also retained its integrity after more than 200 thermal cycles (Supplementary Fig. 18). This may be due to the relatively stable phase interface connected by chemical bonds in natural composites (Supplementary Fig. 19)[36,49].

**Three-dimensional crystallographic orientations**
Using three-dimensional neutron diffraction texture measurements, we analyzed the crystallographic orientation behaviors of the Pre. Er-Fe-B and Tar. Er-Fe-B alloys (Supplementary Fig. 20). The evolutions of the Pre. Er-Fe-B diffraction peaks with the sample stage rotation angle (Ω) collected by the 90° detector (bank 1) are shown in Fig. 3a. The strong crystallographic anisotropy of the ErFe$_{10}$ phase was manifested by the fact that the intensity of the (004) reflection decreased rapidly and disappeared as the sample angle was rotated from 45° to 90°, while the (302) reflection exhibited the opposite trend. The variation of the $(110)_\alpha$ peak intensity was negligible, indicating the weak anisotropy of the $\alpha$ phase therein. Compared to the Pre. Er-Fe-B alloy, the intensities of nearly all the reflections in the Tar. Er-Fe-B, such as $(110)_\alpha$, $(412)_E$, and $(442)_E$, did not change obviously (Fig. 3b), demonstrating their isotropic behavior. To further investigate the interphase orientation relationship, we calculated the neutron pole figures of several characteristic reflections (Fig. 3c, d). The primary $\alpha$ phase in Pre. Er-Fe-B alloy has a weak preferred orientation with $\angle([110]_\alpha, LD) \approx 45°$, whereas the ErFe$_{10}$ phase has a strong fiber texture with [001] // LD. In contrast, both the $\alpha$ and E phases of Tar. Er-Fe-B alloy exhibits

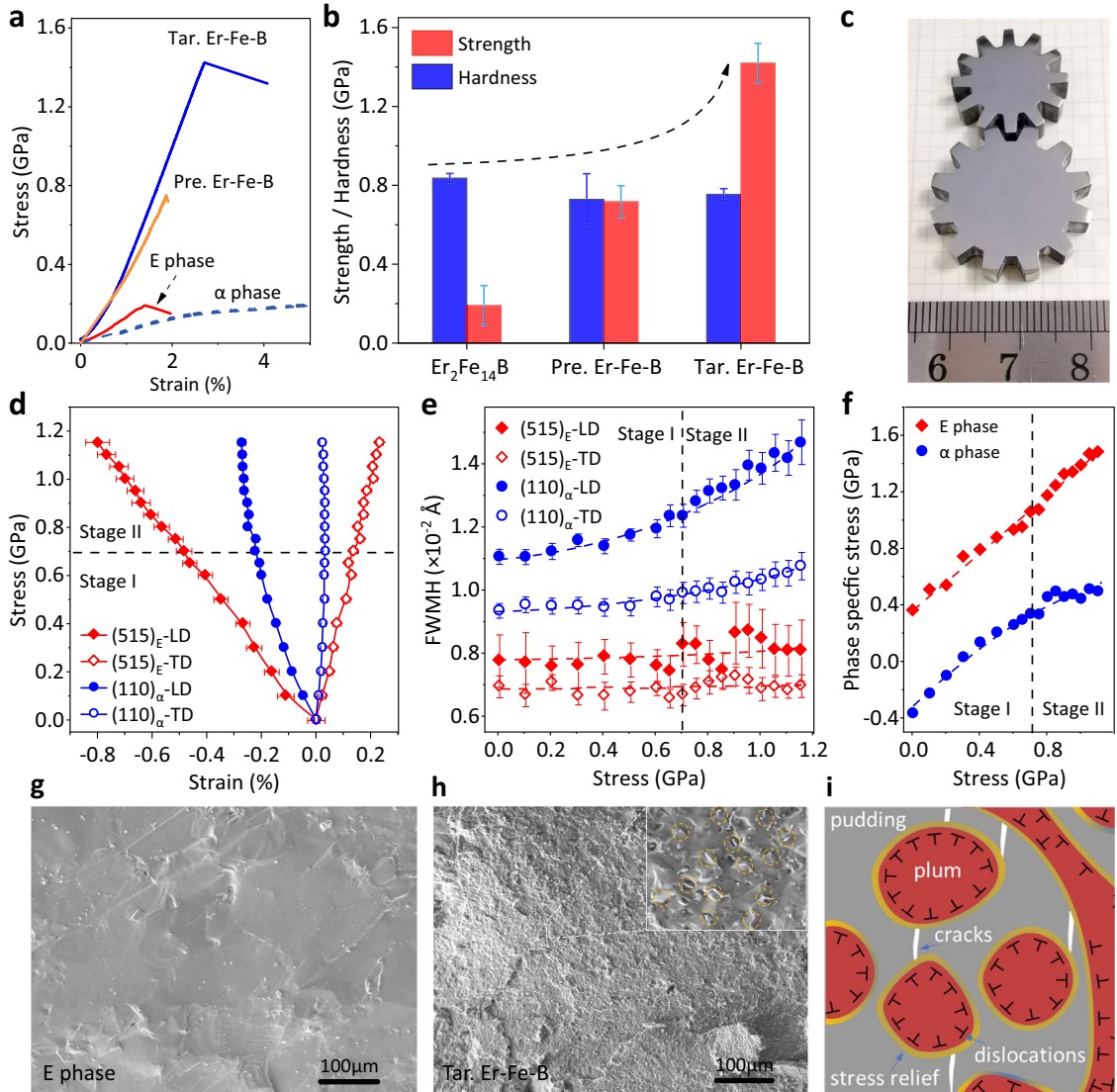

**Fig. 4 | Mechanical performances of the series alloys. a** Engineering compressive stress-strain curves of the ZTE alloy (Tar. Er-Fe-B), Pre. Er-Fe-B, Er₂Fe₁₄B and α-Fe. **b** The compressive strength and Vickers hardness of the as-synthesized alloys. The error bars donate standard deviation. **c** Gears as fabricated in the laboratory. **d, e** Lattice strain $(515)_E$, $(110)_α$ and the corresponding full width at half maximum (FWHM) evolution with the applied stress derived by the single peak fitting. The error bars donate standard deviation. **f** Stress partitioning dependence of the applied stress, Note the tensile stress is depicted as negative. **g, h** The microstructure of the fracture surface of the pure E (**g**) phase and Tar. Er-Fe-B (**h**) alloy. The inset in (**h**) is an enlarged area in which the dimples of the α-phase are indicated by orange circles. **i** The strengthening mechanism of the "plum pudding" microstructure.

uniformly distributed crystallographic orientation relationships (Fig. 3d). The dissimilar crystallographic textures of the Pre. Er-Fe-B and the Tar. Er-Fe-B alloys are attributed to the disordered nucleation and grain growth during reactions[54]. Hence, the BMSR weakens the crystallographic texture of the parent phase and enables the isotropic ZTE property of the Er-Fe-B alloy.

## Mechanical properties

We compared the mechanical properties of the pure E phase, the Pre. Er-Fe-B and the Tar. Er-Fe-B together with the α phase. The E phase (Er₂Fe₁₄B) exhibits intrinsic brittleness with low compressive strength ($δ_{cs}$ = ~191 ± 100 MPa, Fig. 4a), which is a prevalent issue in the majority of ZTE intermetallic compounds. The compressive strength of the Pre. Er-Fe-B is enhanced to $δ_{cs}$ = 800 ± 80 MPa. This exceptional strength is a result of the robustness of the primary α phase at grain boundaries. Intriguingly, the Tar. Er-Fe-B possesses a compressive strength of $δ_{cs}$ = 1.42 ± 0.10 GPa and a toughness of 16.98 ± 1.0 J cm⁻³. Its compressive strength nearly doubles that of the

Pre. Er-Fe-B and is one order of magnitude larger than that of the pure E phase. This indicates that the "plum pudding" microstructure with intragranular α phase precipitations further improves the mechanical properties of the resulting alloy. The hardness of the E phase, the Pre. Er-Fe-B, and the Tar. Er-Fe-B alloys are similar, i.e., 837 ± 23 MPa, 729 ± 129 MPa and 755 ± 29 MPa (Fig. 4b), respectively, but the compressive strength ($δ_{cs}$) is significantly different further demonstrating the dominance of "plum pudding" microstructure on mechanical property improvement. Furthermore, the high elastic modulus ($E$ = 61.47 ± 1.0 GPa) of the Tar. Er- Fe-B reveals its high stiffness. As a result, the present ZTE alloy can be machined into intricately shaped objects such as the gears in Fig. 4c thanks to its improved mechanical properties. The present Tar. Er-Fe-B alloy is distinguished by its comprehensive properties (Supplementary Fig. 21 and Supplementary Table 3).

To understand the mechanism of the improved strength of this dual-phase alloy with "plum pudding" microstructure, in-situ neutron diffraction during loading was conducted (Supplementary Fig. 22). As

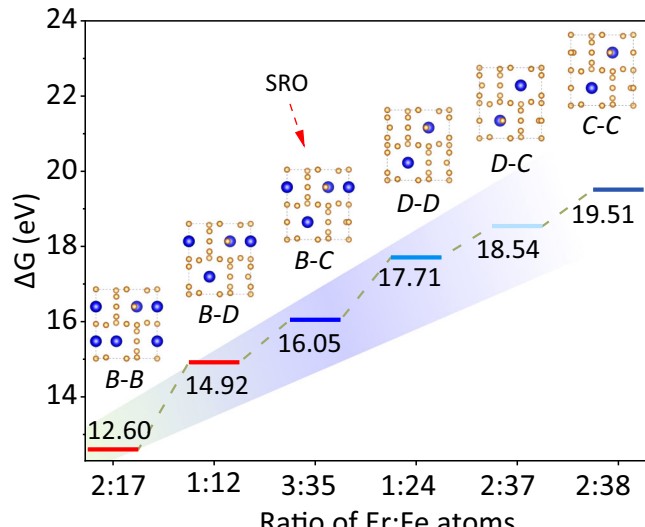

**Fig. 5 | Energy changes ($\Delta G$) correspond to typical structural models.** $B$ represents $Er_{2b}$ site is not substituted, $C$ represents $Er_{2b}$ is replaced by single Fe atoms (usually in the form of Fe-Fe atomic pairs, labeled as $D$ type). As a result, a total of six structural models were established: $B$-$B$ type is stoichiometric $Er_2Fe_{17}$; $B$-$D$ type is $Er_{2b}$ atoms is partly replaced by Fe-Fe pairs; $B$-$C$ is the local chemical ordering discovered in $ErFe_{10}$ (SRO); $D$-$D$ and $C$-$C$ types are $Er_{2b}$ site replaced by Fe-Fe pairs and single Fe atoms, respectively. $D$-$C$ is the $Er_{2b}$ site replaced by Fe-Fe pairs and single Fe atoms hierarchically.

shown in Fig. 4d, there are two deformation stages: (i) cooperative elastic deformation of the two phases ($\delta < 700$ MPa, stage I); (ii) elastic deformation of the E phase and plastic deformation of the $\alpha$ phase ($\delta \geq 700$ MPa, stage II), with the $\alpha$ phase exhibiting work hardening in the plastic stage. The full width at half maximum (FWHM) of $(110)_\alpha$ peak rapidly increases at this stage, indicating that the slip system is activated, and dislocations begin to generate and multiply to transmit stress, thereby reducing and eliminating the stress concentration in grain boundaries and intragranular grains (Fig. 4e). Also, the consistent FWHM of $(515)_E$ along LD and TD reveals the uniform elastic deformation of the E phase, implying the soft $\alpha$ phase plays a key role in the strain delocalization relative to the hard E phase. In addition, Figure 4f demonstrates the phase-specific stress in the E and $\alpha$ phases (Specific calculation in Methods and Supplementary Fig. 23). The E phase, which is a structural cornerstone of the alloy, bears the majority of stress in the whole stage. The $\alpha$ phase bears the load during the synergistic deformation stage and then transmits the stress to the E phase during the plastic deformation stage[38,55,56].

As demonstrated in Fig. 4g, the pure E phase has the typical brittle cleavage fractures, i.e., a smooth and flat surface, and is composed of numerous cleavage planes of roughly equivalent grains. The Tar. Er-Fe-B alloy exemplifies the relative ductile fracture (Fig. 4h and Supplementary Fig. 24), with a bumpy surface and spectacular dimples of various shapes that are uniformly and densely distributed in the microstructure, indicating that the phase has undergone significant plastic deformation. They are compressed, twisted, and sheared to be extracted from the matrix. During the loading process, a large amount of energy is absorbed, the internal stress is released at the phase boundaries, and the propagation of E-phase cracks is inhibited, resulting in an improved ability of the E phase to bear load (Fig. 4i). Thus, the precise engineering of the grain boundaries and intragranular grains in this dual-phase alloy to create a "plum pudding" microstructure can significantly enhance its strength and toughness of the control of grain boundary and intragranular grain engineering to architect "plum pudding" microstructure can greatly improve strength and toughness[36,49].

## Mechanisms for the boron-migration-mediated solid-state reaction (BMSR)

Finally, we discuss the key factors in the BMSR of the current system. As presented above, upon the selective migration of boron atoms from the grain boundary to the precursor grains during the prolonged annealing, the initial 2:17-type $ErFe_{10}$ matrix transforms to a 2:14:1-type E phase, and the intergranular $\alpha$ phase re-precipitates and homogeneously disperses into the tetragonal E phase matrix, forming the Tar. Er-Fe-B alloy. Apparently, the slow selective migration of boron atoms from the grain boundaries to the $ErFe_{10}$ matrix is beneficial for the formation of such a homogenous "plum pudding" microstructure. The sluggish diffusion of light atoms (boron and carbon elements) has also been observed in other metal materials[15,51,56]; Furthermore, in the Pre. Er-Fe-B alloy, we identified six potential chemical configurations (Supplementary Fig. 25) based on the substitution of $Er_{2b}$ position ($B$ types) by Fe-Fe pair ($D$ types) and single Fe atoms ($C$ types)[21,57]. According to the density functional theory (DFT) calculations, the chemical short-range order ($B$-$C$ types, SRO) has a higher binding energy ($\Delta G$) than the stoichiometric $Er_2Fe_{17}$ (Fig. 5), indicating its structural metastability. The metastability of the special chemical SRO in the precursor could reduce the energy barrier of its reaction with the migrated boron and may serve as a nucleation site to promote the BMSR.

In a summary, a multi-property profile stable ZTE dual-phase alloy (Tar. Er-Fe-B) is achieved by employing boron-migration-mediated solid-state reaction (BMSR), which boosts isotropic ZTE (three-dimensional stability), wide temperature window (100 – 550 K, $\Delta T = 450$ K), combined high strength-stiffness ($\delta_{cs} = 1.44 \pm 0.10$ GPa and $E = 61.47 \pm 1.0$ GPa) and robust thermal shock resistance. In the $Er_2Fe_{14}B$ matrix with a negative thermal expansion, the submicron $\alpha$-Fe with positive thermal expansion precipitates in both ingrain and intergrain forms. Chemical short-range order (SRO) and the sluggish selective diffusion of boron atoms are crucial to the efficient occurrence of the BMSR. This reaction disrupts the local crystallographic texture of the Pre. Er-Fe-B phase, resulting in isotropic ZTE performance. BMSR also leads to the formation of a unique "plum pudding" microstructure, which can enhance the strength and toughness of the resulting Tar. Er-Fe-B. By manipulating microstructures, we anticipate more ZTE alloys and high-performance functional materials could be generated.

## Methods
### Sample preparation
The samples of $Er_2Fe_{14+x}B_{1+0.07x}$ ($x = 0$, 2.5, 5, 12.5, 22.5) were prepared by arc melting with Er, Fe and B elements ( > 99.9% purity) under high purity argon atmosphere. The samples were turned over and melted four times to ensure homogeneity. Then, the sample was followed by annealing in two steps: (i) high temperature (1473 K) in an argon atmosphere for about 24 hours (Precursor alloy); (ii) low temperature (1323 K) for about 120 hours and quenched in liquid nitrogen (Target alloy).

### Structural and electron microscopy characterization
The microstructure analysis was measured by electro-probe microanalyzer backscattering electron (EPMA-BSE) spectrum (SHIMADZU 1720) equipped with wave-length dispersive spectrometer analysis (WDS) to quantitatively determine the phase composition. The brightfield images, SAED, and high-resolution transmission electron microscopy (HRTEM) were conducted at FEI Tecnai F30 transmission electron microscopy (TEM). The HAADF-STEM and ABF-STEM images were obtained on an aberration-corrected TEI Tecnai ETEM, JEM-ARM 200 F. The surface of fracture microstructure orientation of the samples was measured by scanning electron microscope (SEM, Zeiss Geminisem 500). SEM and EPMA samples were polished down to the 2000-grit SiC paper and then polished with a metal polishing agent.

TEM samples were mechanically ground to 50 μm thickness, and then twin-jet electropolished using $H_2SO_4$ (10%) and $CH_4O$ (90%) solution under −30 °C. HAADF-STEM samples were fabricated by focused ion beam (FIB) into 50 nm thick slices.

## Mechanical properties measurements and dilatometer thermal expansion

The room-temperature strain-stress curves were measured using a CMT4105 universal electronic compressive testing machine with a Φ 6 × 8 mm cylinder and an initial strain rate of 0.25 mm/min. The Vickers hardness was measured by a Vickers diamond indenter (FALCON 507, INNOVATEST, Netherlands) with a load of 1.96 N for 10 s. The dilatometer thermal expansion was tested by an advanced thermodilatometer (NETZSCH DIL402). The coefficient of thermal expansion of $\alpha_l$ was calculated by the Eq. (1):

$$\alpha_l = \frac{dL}{L_0}/(dT) \tag{1}$$

In the formula $dL/L_0 = (L_1-L_0)/L_0$, $dT = T_1-T_0$; It is defined as the ratio of the length change ($dL/L_0$) to the temperature interval within the specified temperature range ($dT$).

## Crystal structure and crystallographic texture characterization

Single crystal diffraction was determined by X-ray diffraction analysis at 150 K using an Oxford Diffraction Gemini E system with Mo Kα radiation, $\lambda = 0.71073$ Å. The three-dimensional crystallographic texture and in-situ loading study by neutron diffraction were carried out at the VULCAN beamline (BL-7) in Oak Ridge National Laboratory (ORNL), USA. The temperature dependence of synchrotron X-ray diffraction (SXRD) of the samples was collected at the BL44B2 beamline in SPring-8 ($\lambda = 0.45$ Å), Japan.

## Magnetization measurements

The magnetization measurements were measured by a physical property measurement system (PPMS) of Quantum Design with the vibrating sample magnetometer (VSM), which is cooled by liquid helium

## Lattice thermal expansion of the Er-Fe-B dual alloy

Due to the homogeneous microstructure with isotropic crystallographic texture. The lattice thermal expansion of dual-phase alloy was calculated as Eq. (2) and (3):

$$\alpha_l = \frac{(\Sigma a_1 - \Sigma a_0)}{3\Sigma a_0}/(T_0 - T_0) \tag{2}$$

$$\Sigma_a = mol._{\alpha}\% \times V_\alpha + mol._E\% \times V_E \tag{3}$$

where $\alpha_l$ is the apparent lattice thermal expansion; $mol._{\alpha}\%$ and $mol._E\%$ are molar fractions of the α and E phase determined by the results of SXRD data.

## Lattice strain under loading calculations

The lattice strain of the specific (h k l) reflections during the loading was determined by the single peak fitting method. The lattice strain was calculated by following Eq. (4)

$$\text{Strain} = \frac{(d_1 - d_0)}{d_0} \times 100\% \tag{4}$$

Here, $d_1$ and $d_O$ represent the interplanar crystal spacing of the (h k l) crystal plane after and before loading, respectively. For average lattice strain ($\varepsilon_i$), the $d_1$ and $d_O$ are replaced by the unit cell parameters ($a_1$ and $a_O$).

## Thermal residual expansion and phase-specific stress calculation

Owing to the mismatch in thermal expansion of the two phases, the thermal residual stress was evaluated by Eq. (5):

$$\sigma_r = \int_{RT}^{T_C} \Delta\alpha \times E \, dT \tag{5}$$

where $\Delta\alpha_l$ is the difference in CTE ($\Delta\alpha_l = \alpha_{l,\,\alpha} - \alpha_{l,\,E}$), $E$ is elastic modulus determined by engineering stress-strain curves, RT is room temperature, $T_C$ is the curie temperature.

The phase-specific stress was calculated by following Eq. (6):

$$\sigma_i = \frac{E_i}{(1+v_i)(1-2v_i)} \times \{(1-v_i) \times \varepsilon_{i,11} + v_i \times (\varepsilon_{i,22} + \varepsilon_{i,33})\} + \sigma_r \tag{6}$$

where $i$ stands for the α and E phase, $\sigma_i$ is the stress in the loading direction, $E_i$ is the diffraction elastic modulus, $v_i$ is the Poisson's ratio, $\sigma_r$ is the thermal residual stress, $\varepsilon_{i,\,11}$ is the lattice strain in LD, $\varepsilon_{i,\,22}$ and $\varepsilon_{i,\,33}$ are the lattice strains in TD and ND, respectively. The $\varepsilon_{i,\,22} = \varepsilon_{i,\,33}$ and can be measured by TD.

## Density functional theory (DFT)

DFT calculations were conducted by Vienna ab initio simulation package (VASP) with the Perdew–Burke–Ernzerhof (PBE)-generalized exchange and correlation energy. The binding energy ($\Delta G$) is calculated as the Eq. (7):

$$\Delta G = E(Er_n Fe_m) - nE(Er) - mE(Fe) \tag{7}$$

## Data availability

The data that support the findings of this study are available from the corresponding authors upon request.

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

## Acknowledgements

This research was supported by the National Key R&D Program of China (2020YFA0406202) (X.R.X.), the National Natural Science Foundation of China (22090042 and 21971009) (X.R.X.), Guangxi BaGui Scholars Special Funding, and the Fundamental Research Funds for the Central Universities, China (FRF-IDRY-GD21-03 and GJRC003) (K.L.). The synchrotron radiation experiments were performed at the BL44B2 of SPring-8 with the approval of the Japan Synchrotron Radiation Research Institute (JASRI) (Proposal No. 2019A1378, 2018B1515); Neutron diffraction work was carried out at the Spallation Neutron Source (SNS) (Proposal No. 2020B26069), which is the U.S. Department of Energy (DOE) user facility at the Oak Ridge National Laboratory, sponsored by the Scientific User Facilities Division, Office of Basic Energy Sciences. We thank Dr. Masato Hoshino for the support in high-energy X-ray CT, JASRI, SPring-8, Japan.

## Author contributions

X.X., K.L. and C.Y. conceived the idea of the work and supervised the project. C.Y. synthesized the alloys. C.Y. and W. L. carried out the main experiments. C.Y., L.K. and K.K. analyzed the SXRD data. Y.Ca. helped the measurements of magnetism. L.C. processed the single crystal diffraction data. X.K., J.D. and Q.L. analyzed the thermal expansion results. Q.Z., L.Y. and L.G. conducted the TEM measurements. S.J. helped analyze the TEM results. X.C. carried out the theoretical calculation. H.W., K.A., Y.Ch. and D.Y. analyzed the in-situ neutron diffraction results. All authors discussed the results and commented on the manuscript.

## Competing interests

The authors declare no competing interests.
