## [Peer Review File · Nature Communications]

Superior zero thermal expansion dual-phase alloy via boron-migration mediated solid-state reactionREVIEWER COMMENTS

Reviewer #1 (Remarks to the Author):

This work reports a new composite alloy with wide operating temperature windows, high strength and low thermal expansion. Generally it is written in a good way taking high-tech characterization methods and in-depth discussions. Several points need to consider before its publication:

- (1) p5. in Fig.1a, explain the lattice sites in more details, as there is no information in its corresponding caption. More importantly, give the background of 'SRO' and why is it relative to the solid reaction.
- (2) p7. line 149, how to deduce the conclusion of 'stable robust chemical interface' from Supplementary Fig.14 of HRTEM?
- (3) Have you assessed the tensile performance besides the currently conducted compressive mode?
- (4) Supplementary P3, marked in Figure S3, should as S2.

Reviewer #2 (Remarks to the Author):

The authors report a novel dual-phase ZTE alloy exhibiting isotropic zero thermal expansion over a wide temperature range together with excellent mechanical properties and thermal stability. It was designed by a "plum pudding" architecture via a solid-state reaction. This method is interesting and impressive for researchers focusing on ZTE materials and microstructural design. The results are solid to support their claims. It should be suitable for a publication in NM after the following revisions:

- (1) A novel architecture is achieved by a solid-state reaction in the selected alloy system. Would the design method be applied to other alloy systems?
- (2) The reported new phase in the precursor alloy is out of stoichiometric ratio. What's the effect on thermal expansion or other physical properties?
- (3) What is D in Fig. 4b?
- (4) The in-situ NPD data in Fig. 4f would show that the dual phases are internally stressed before loading. How did it come about? Does it affect mechanical properties ?
- (5) The FWHM of these peaks in Fig. 4e would be replaced by FWHM/d.

Reviewer #3 (Remarks to the Author):

The present manuscript reports an alloy with superior zero thermal expansion within a wide temperature range (100 - 500 K). The authors attributed these superior properties to the "plum pudding" structure, which can be explained by a B migration mediated solid state reaction. In the $\text{Er}_2\text{Fe}_{14}\text{B}$ alloy, the $\alpha\text{-Fe}$ particles with diameter about 200 nm dispersed uniformly in $\text{Er}_2\text{Fe}_{14}\text{B}$ matrix. The $\text{Er}_2\text{Fe}_{14}\text{B}$ matrix shows negative thermal expansion with increasing the temperature, counteracting the positive thermal expansion of $\alpha\text{-Fe}$. In addition, the texture in pre. Er-Fe-B alloy disappears in tar. Er-Fe-B alloy, enabling isotropic behavior under thermal conditions. The results are

interesting and worth publishing after the following questions being answered properly.

1. The atoms with different colors should be labelled clearly in Fig. 1, and the atom arrangement of E phase along $\langle 100 \rangle$ direction should be presented. The SEM and TEM specimen preparation method should also be given in detail. Are Fig. 1b and f SE images or BSE images? The appearance of wavy structure in Fig. 1i was attributed to the mutual extrusion during the nucleation and growth of α -Fe grains. But I think the stress can be released by other easier way, such as grain rotation, grain boundary or interfacial structure relax. Can these wavy structures be frequently observed in another specimen?
2. The peaks in the XRD profiles should be labelled in a clearer way. From Fig. S2, the diameter of $\text{Er}_2\text{Fe}_{14}\text{B}$ matrix is approximately 500 nm, which is significantly larger than those in Fig. 1 and Fig. S1 (~ 200 nm). Are they from two specimens or two regions of the same specimen?
3. As the authors claimed, the volume fraction of phase or B content can tune the thermal expansion performance of the Er-Fe-B alloy. Can you show their relationship in a quantitative way? Or can you measure the volume fraction of phases in samples with different B content?
4. What is α l and its relation with d l/l? The zero line in each subfigure in Fig. 2 should be marked.
5. Can the authors show the microstructure evolution (α -Fe precipitation process) during thermal annealing? In addition, if 2:17 type structure has lower energy, why the B-C SRO appear in Fig. 1. What is the relation between B-C SRO and $\text{Er}_2\text{Fe}_{17}$ phase? More importantly, how B modifies the structure of ErFe_{10} to form $\text{Er}_2\text{Fe}_{14}\text{B}$? This should be discussed in more detail.

Reviewer #1 (Remarks to the Author):

This work reports a new composite alloy with wide operating temperature windows, high strength and low thermal expansion. Generally it is written in a good way taking high-tech characterization methods and in-depth discussions. Several points need to consider before its publication:

Reply: Thanks for your interest in our current work and for recommending acceptance for publication in *Nature Communications*. We also appreciate your insightful comments and constructive suggestions. Following these comments/suggestions, we have (i) added detailed structural information and discussion of ErFe_{10} and SRO; (ii) added the background of SRO; (iii) calculated the binding energy of the corresponding structure by density functional theory (DFT) to reveal the relationship of SRO and solid reaction. (iv) discussed the case of phase interface binding and (v) revised some mistakes. *Based on these results, we have completely revised the manuscript. The detailed corrections are listed below.*

Comments 1: *p5. in Fig.1a, explain the lattice sites in more details, as there is no information in its corresponding caption. More importantly, give the background of 'SRO' and why is it relative to the solid reaction.*

Reply: Thanks for your constructive comments and suggestions. (i) We have added the detailed structural information and discussion in the revised manuscript: “*The metastable ErFe_{10} is a new phase discovered here, which possesses a 2:17-type topological structure ($P6_3/mmc$) with seven Wyckoff sites (Er_{2b} , Er_{2d} , Fe_{4f} , Fe_{6g} , Fe_{12j} , Fe_{12k} and Fe_{4e}). These excess Fe atoms mainly replace the Er_{2b} sites disorderly in the manner of Fe-Fe pairs to form Fe_{4e} lattice sites (A types, Fe-Fe pairs, and Er atoms disordered substitution, Fig. 1a), as confirmed by synchrotron X-ray diffraction.*” Besides, we have added the detailed crystal model and Table in the revised supporting information (Fig. R1 and Table R1).

Fig. R1 The crystal structure model of ErFe_{10} .

Table R1 The long-range crystal structure information of ErFe_{10} determined by SXR D.

Phase	Atoms	x	y	z	Occ.	Site	Sym.
	Er ₁	0.00000	0.00000	0.25000	0.772	2b	-6m2
	Er ₂	0.33333	0.66667	0.75000	1.000	2d	-6m2
	Fe ₁	0.33333	0.66667	0.10676	1.000	4f	3m.
ErFe_{10}	Fe ₂	0.50000	0.00000	0.00000	1.000	6g	.2/m.
	Fe ₃	0.33415	0.95614	0.25000	1.000	12j	$m..$
	Fe ₄	0.16667	0.33296	0.97899	1.000	12k	.m.
	Fe ₅	0.00000	0.00000	0.89400	0.231	4e	3m.

(ii) Compared with general chemically disordered occupancy, atoms tend to attract or repel each other in the local area (nanoscale), resulting in the

preferential choice of avoidance or aggregation among neighboring atoms, which may lead to a local regular lattice site occupancy, that is, chemical short-range order (SRO). In the present work, we have determined the average structure of ErFe_{10} to be $\text{Er}_2\text{Fe}_{17}$ -type structure ($P6_3/mmc$) by synchrotron radiation X-ray diffraction (SXRD). The excess Fe atoms mainly replace the Er_{2b} sites disorderly in the manner of Fe-Fe pairs to form Fe_{4e} lattice sites. But we are still curious whether there is a difference between the local structure and the average structure. As a result, we further conducted the aberration-corrected STEM experiment. The high-angle annular dark-field (HAADF) and annular bright-field (ABF) images evidence a chemical short-range order (SRO), where the single Fe atom replaces the Er_{2b} site hierarchically to form B-C-B-C (C: Er_{2b} replaced by a single Fe atom; B: Er_{2b} not replaced) layers along the c axis as shown in Fig. R2. The Er_{2b} site hierarchically occupied by a single Fe atom in the local area is different from the standard $\text{Er}_2\text{Fe}_{17}$ structure and the average ErFe_{10} structure in long ranges, that is the chemical short-range order (SRO), as shown in Fig. R3.

Fig. R2 HAADF images of the ErFe_{10} phase along $[100]$ zone axis.

Fig. R3 The structure model of the ErFe₁₀ phase.

(iii) The B-C-B-C-type SRO was observed in the ErFe₁₀ phase for the first time, which was different from the average structure. And we enumerated all possible chemical configurations of this hexagonal lattice (Fig. R4): **(1)** B-B type, ground state 2:17 compound; **(2)** B-D type, Er₂ site was replaced by Fe-Fe pair; **(3)** B-C type, Er₂ site was replaced by single Fe atom (short-range order, SRO); **(4)** D-D type, Er₁, and Er₂ site were replaced by Fe-Fe pairs; **(5)** D-C type, Er₁ site was replaced by Fe-Fe pair and Er₂ was replaced by single Fe atom; **(6)** C-C type, Er₁, and Er₂ site were replaced by single Fe atoms. **We further calculated the binding energy of the corresponding structure by density functional theory (DFT), and the results showed that the Fe-Fe pair (B-D type) and single Fe atom (B-C type, SRO) replaced rare earth site has a higher binding energy (ΔG) compared with the ground state Er₂Fe₁₇, indicating that the metastability of its structure (Fig. R5).** That is to say, the ErFe₁₀ is metastable. The other three types D-D, D-C, and C-C have higher binding energy and lower stability. It has not been directly observed experimentally and the structure may collapse at room temperature. As a result, the B-C type short-range chemical order (SRO) reduces the energy barrier of

the solid-phase reaction due to its metastability and can be used as a nucleation site for the reaction to promote the solid-phase reaction so a "plum pudding" structure can be built.

We have added the above discussion and corresponding results in the revised manuscript and support information.

Fig. R4 all possible chemical configurations of this hexagonal lattice.

Fig. R5 Energy changes (ΔG) corresponding to typical structural models.

Comments 2: p7. line 149, how to deduce the conclusion of 'stable robust chemical interface' from Supplementary Fig.14 of HRTEM?

Reply: Thanks for your insightful comment. In the field of zero thermal expansion composites, weak phase interface bonding may lead to fatigue failure of the material. This is due to the thermal expansion mismatch of the dual-phase alloys. And it is generally believed that the interface obtained by traditional powder solid-state sintering will have inevitable element oxidation and segregation, thus deteriorating its interfacial bonding, such as $\text{ZrW}_2\text{O}_8/\text{Al-Si}^1$ and La-Fe-Co-Si/Cu^2 . In the present work, the dual-phase alloys were “natural composite”. And the dilatometer thermal expansion remained constant after 200 thermal cycles (Fig. R6), which confirmed its cycles thermal stability. Besides, we conducted the electro-probe micro-analyzer (EPMA) measurements after 200 thermal cycles. No observable microcracks emerge at the interface (Fig. R7). And further phase interface structure collected by chemical bonds can support the relative stability (Fig. R8).

In this version, to make it more reasonable, we changed it to “*This may be due to the relatively stable phase interface connected by chemical bonds in natural composites (Supplementary Fig. 14)*”. And we have cited relative literature on chemical interfaces ((a) Lu K., Lu L., Suresh S. *Strengthening materials by engineering coherent internal boundaries at the nanoscale [J]. Science, 2009, 324(5925): 349-352*); (b) Ding R., Yao Y., Sun B., et al. *Chemical boundary engineering: a new route toward lean, ultrastrong yet ductile steels [J]. Science Advances, 2020, 6(13): eaay1430.*)

Fig. R6 The dilatometer thermal expansions of Tar. Er-Fe-B alloy in the 1st, 100th, and 200th cycles. Insert the cyclic thermal shock experiment that undergoes a thermal shock from 77 K to 600 K.

Fig. R7 The microstructure of the Tar. Er-Fe-B after 200th thermal cycles.

Fig. R8 The two typical phase interfaces of the Tar. Er-Fe-B alloy.

Comment 3: *Have you assessed the tensile performance besides the currently conducted compressive mode?*

Reply: We also look forward to achieving tensile plasticity, but it is hard to

achieve this goal in this work.

Comment 4: *Supplementary P3, marked in Figure S3, should as S2.*

Reply: Sorry for the mistake. We have revised it in the present manuscript.

Reviewer #2 (Remarks to the Author):

The authors report a novel dual-phase ZTE alloy exhibiting isotropic zero thermal expansion over a wide temperature range together with excellent mechanical properties and thermal stability. It was designed by a “plum pudding” architecture via a solid-state reaction. This method is interesting and impressive for researchers focusing on ZTE materials and microstructural design. The results are solid to support their claims. It should be suitable for publication in NC after the following revisions:

Reply: We appreciate your interest in our recent study and your suggestion that it be published in *Nature Communications*. In addition, we appreciate your insightful remarks and constructive suggestions. In response to these comments/suggestions, we have (i) conducted magnetic measurements on the precursor Er-Fe-B alloy, which revealed that its thermal expansion and magnetic properties have no significant effect; (ii) added the details of calculating thermal residual stress σ_r and (iii) checked the mistake in the present version. We have revised the entire manuscript. The corrections are enumerated in detail below.

Comment 1: *A novel architecture is achieved by a solid-state reaction in the selected alloy system. Would the design method be applied to other alloy systems?*

Reply: In this work, the idea of “boron-migration-mediated solid-state reaction” originated from the “solid solution re-precipitation” in materials science, by constructing a supersaturated solid solution under high-temperature conditions, and then re-precipitating under low-temperature conditions. The method of boron-migration-mediated solid-state reaction may also be used in other alloy systems under reasonable design. We will continue to study the general applicability of the strategy in the follow-up work.

Comment 2: *The reported new phase in the precursor alloy is out of stoichiometric ratio. What's the effect on thermal expansion or other physical properties?*

Reply: The new ErFe_{10} phase possesses a 2:17-type topological structure ($P6_3/mmc$) but with 16.7 % excess Fe atoms. These excess Fe atoms mainly replace the Er_{2b} sites disorderly. Besides, we also observed a single Fe atom replacing the rare earth Er_{2b} site in a chemical short-range order (SRO) manner (B-C-B-C) for the first time. Collectively, the anti-site occupancy on crystal structures leads to deviations from the standard stoichiometric ratio. We have conducted linear thermal expansion and magnetic measurements. The ErFe_{10} phase shows negative thermal expansion (NTE) in the temperature range of 100 – 315 K (Fig. R9 a), which is consistent with a macroscopic ferrimagnetic to paramagnetic transition (Fig. R9 b-c). Both thermal expansion and Curie temperature ($T_C = 315$ K) did not change significantly from previously reported $\text{Er}_2\text{Fe}_{17}$ phases³. As a result, the excess Fe atoms have no obvious effect on its thermal expansion and magnetic properties.

Fig. R9 The thermal expansion and magnetization curves of Pre. Er-Fe-B

alloy. a The dilatometer thermal expansion of the Pre. Er-Fe-B (ZTE composition); **b-c** The magnetization curves of the Pre. Er-Fe-B (ZTE composition).

Comment 3: *What is D in Fig. 4b?*

Reply: Sorry for mistake, we removed it in the revised manuscript.

Comment 4: *The in-situ NPD data in Fig. 4f would show that the dual phases are internally stressed before loading. How did it come about? Does it affect mechanical properties?*

Reply: The internal stress originates from the thermal expansion mismatch of the two phases, the α phase shows positive thermal expansion ($\alpha_l = 16.5 \times 10^{-6} \text{K}^{-1}$), and E phase shows negative thermal expansion ($\alpha_l = -6.6 \times 10^{-6} \text{K}^{-1}$), the thermal residual stress was evaluated by the formula:

$$\sigma_r = \int_{RT}^{T_c} \Delta\alpha \times E dT$$

where $\Delta\alpha$ is the difference in CTE ($\Delta\alpha_l = \alpha_{l,\alpha} - \alpha_{l,E}$), E is elastic modulus determined by engineering stress-strain curves, RT is room temperature, and T_c is the curie temperature. The thermal residual stress $\sigma_r = 363.6 \text{ MPa}$, which affects the mechanical behavior of the two phases. The result reveals a tensile residual stress in the α phase and a compressive residual stress in the E phase. Subsequently, during the compression test, it has to first overcome the tensile residual stress in α before putting its lattice into compression. That's why the α phase yielded at $\delta = 600 \text{ MPa}$.

Comment 5: *The FWHM of these peaks in Fig. 4e would be replaced by FWHM/d.*

Reply: We revised it in the present manuscript.

Reviewer #3 (Remarks to the Author):

The present manuscript reports an alloy with superior zero thermal expansion within a wide temperature range (100 - 500 K). The authors attributed these superior properties to the "plum pudding" structure, which can be explained by a B migration mediated solid state reaction. In the tar. Er₂Fe₁₄B alloy, the alpha-Fe particles with diameter about 200 nm dispersed uniformly in Er₂Fe₁₄B matrix. The Er₂Fe₁₄B matrix shows negative thermal expansion with increasing the temperature, counteracting the positive thermal expansion of alpha-Fe. In addition, the texture in Pre. Er-Fe-B alloy disappears in Tar. Er-Fe-B alloy, enabling isotropic behavior under thermal conditions. The results are interesting and worth publishing after the following questions being answered properly.

Reply: We are grateful to the reviewer for recognizing the potential value of our work and recommending publication in *Nature Communications*. We value the reviewer's insightful comments and constructive suggestions as well. In response to these comments/suggestions, we have (i) added experimental details for SEM/TEM in methods; (ii) conducted TEM experiments to confirm the origin of the wavy structure of α phase; (iii) supplemented morphology data for different sizes of α phase to illustrate the range of grain sizes; (iv) conducted room temperature X-ray powder diffraction (XRD) and synchrotron radiation X-ray diffraction (SXRD) to confirm the quantitative relationship between mass fraction and thermal expansion; (v) supplemented the microstructural evolution in at different stages. Based on the results of these experiments, we revised the manuscript extensively. Below is a summary of the specific corrections.

Comment 1: *The atoms with different colors should be labelled clearly in Fig. 1, and the atom arrangement of E phase along <100> direction should be presented. The SEM and TEM specimen preparation method should also be given in detail. Are Fig. 1b and f SE images or BSE images? The appearance of wavy structure in Fig. 1i was attributed to the mutual extrusion during the nucleation and growth of alpha-Fe grains. But I think the stress can be released*

by other easier way, such as grain rotation, grain boundary or interfacial structure relax. Can these wavy structures be frequently observed in another specimen?

Reply: Thanks for your constructive comment and suggestion. (i) We have added the labels of Er, Fe, and B atoms in the revised manuscript. (ii) We have added the structure model of the E phase along $\langle 100 \rangle$ in the present manuscript. (iii) We have added experimental details for SEM/TEM in methods. (iv) Fig. 1b and f are BSE images based on electro-probe microanalyzer backscattering electron (EPMA-BSE).

(v) For the wavy structure, the sample was prepared by a focused ion beam (FIB), so this sample has only a thin area with a large field of view (Fig. R10 a). The morphology and selected electron diffraction show that the α phase is polycrystalline on the nanometer scale (Fig. R10 b-c). That is to say, the α phase with the size of 0.1-10 μm was composed of multiple small nano-sized grains. Further high-resolution transmission electron microscopy (HRTEM) indicated that there are regular lattices and wavy lattice fringes of α phase in the sample, see Fig. R10 d-f. Due to the observed microscopic polycrystalline nanostructure and wave structure, we concluded that this was related to the mutual extrusion during the nucleation and growth of α grains nucleation and growth.

In this version, to further verify whether this wave structure is common in the α phase, we have re-prepared the sample with twin-jet electropolished and have carried out a TEM test. Three independent α phase grains were observed, and each grain selected two independent regions (Fig. R11 a, d, and g). Strangely, the α phase maintained a regular lattice arrangement, and no wave-like structure was observed (Fig. R11). Based on the above results, we asked Prof. Jiang (USTB, China) for advice. After careful discussion, we believed that this wave structure may be caused by the distortion of the α phase during the preparation process by the focused ion beam (FIB) and it was not the intrinsic information of the α phase.

Therefore, we have deleted the wave structures, and replaced them with the latest data in the revised manuscript (Fig. R12).

Fig. R10 The TEM data of Tar. Er-Fe-B alloy (ZTE composition). **a**, A typical E/ α phase interface prepared by a focused ion beam (FIB). **b**, The microstructure of the polycrystalline nanostructure. **c**, The select area electron diffraction (SAED) of the area marked by the red box in (**c**). **d-f**, HRTEM in different regions.

Fig. R11 The TEM data of the α phase. **a, d, g** Three independent α phase thin areas. Insert the selected area of electron diffraction (SAED). **b-c, e-f, h-i** The HRTEM results correspond to **a, d, and g** areas, respectively.

Fig. R12 The revised Fig. 1

Comment 2: *The peaks in the XRD profiles should be labelled in a clearer way. From Fig. S2, the diameter of $\text{Er}_2\text{Fe}_{14}\text{B}$ matrix is approximately 500 nm, which is significantly larger than those in Fig. 1 and Fig. S1 (~200 nm). Are they from two specimens or two regions of the same specimen?*

Reply: (i) We have modified the XRD profiles carefully in the revised version. (ii) Sorry for misleading you. The $\text{Er}_2\text{Fe}_{14}\text{B}$ matrix is approximately 200 μm . And the size of the α phase is about 0.1 ~ 10 μm (Fig. R13). We're guessing you care about the size of the α phase. It is uniform on the micron scale (Fig. R13 a), but has a large span at the nanoscale (100 – 1000 nm), as shown in Fig. R13 b-f. Large grains are formed by the aggregation of small nano-polycrystals (Fig. R13 c), and there are also some small-sized single-crystal nanoparticles (Fig.13 e-f). We believed that the reason for this phenomenon lies in the different order of kinetics during the solid phase reaction.

We have added the discussion and corresponding result in the revised

manuscript and supporting information.

Fig. R13 The microstructure of the α phase. **a** The enlarged electro-probe micro-analyzer (EPMA) image of the Tar. Er-Fe-B alloy. **b-f** The transmission electron microscopy (TEM) morphology of precipitated α phase.

Comment 3: *As the authors claimed, the volume fraction of phase or B content can tune the thermal expansion performance of the Er-Fe-B alloy. Can you show their relationship in a quantitative way? Or can you measure the volume fraction of phases in samples with different B content?*

Reply: As you noticed, the matrix of the $\text{Er}_2\text{Fe}_{14}\text{B}$ phase exhibits a negative thermal expansion (NTE), whereas the α phase shows a positive thermal expansion. Consequently, we adjusted the thermal expansion by introducing more α phase ($\text{Er}_2\text{Fe}_{14+x}\text{B}_{1+0.07x}$; $x = 0.0, 2.5, 5.0, 12.5,$ and 22.5). And the excess of a small amount of B atoms was to ensure that the solid phase reaction fully occurs.

In this version, we have conducted the room temperature X-ray diffraction (XRD) of the $x = 0.0, 2.5$ and 5.0 series alloys. However, because the Fe content

of $x = 12.5$ and $x = 22.5$ is too high, the mechanical properties are pretty good, it is difficult to get high quality diffraction peaks on laboratory diffractometers (XRD). As a result, we have supplemented the synchrotron radiation X-ray diffraction (SXRD) measurement of $x = 12.5$ and $x = 22.5$ compositions (Japan, Spring-8, BL44B2). Base on XRD and SXRD data, we have obtained the contents of α phase (mass %) based on the Rietveld refinement (Fig. R13 b-d): $4.67 \pm 0.07 \%$ ($x = 0$), $10.91 \pm 0.1 \%$ ($x = 2.5$), $17.43 \pm 0.04 \%$ ($x = 5$), $42.51 \pm 0.16 \%$ ($x = 12.5$) and $56.71 \pm 0.16 \%$ ($x = 22.5$).

Fig. R14 The X-ray diffraction profiles. a The Rietveld refinement of $x = 0$, $x = 2.5$ and $x = 5$ compositions (XRD, $\lambda = 1.45 \text{ \AA}$). **b** The Rietveld refinement of $x = 12.5$ and $x = 22.5$ compositions (SXRD, $\lambda = 0.70 \text{ \AA}$).

Then we summarized composition ($x = 0, 2.5, 5, 12.5,$ and 22.5) versus the coefficient of thermal expansion (CTE) and contents of α phase (mass %) in Fig. R15. It shows that with the increase of x , the coefficient of thermal

expansion increases continuously, and the content of the α phase is consistent with it. We have supplemented the above discussions in the revised manuscript.

Fig. R15 The comparison of compositions versus CTE and contents of α phase.

Comment 4: *What is α_i and its relation with dL/L ? The zero line in each subfigure in Fig. 2 should be marked.*

Reply: (i) The coefficient of thermal expansion of α_i was calculated by the formula:

$$\alpha_i = \frac{dL}{L_0} / (dT)$$

In the formula $dL/L_0 = (L_1 - L_0)/L_0$, $dT = T_1 - T_0$; It is defined as the ratio of the length change (dL/L_0) to the temperature interval within the specified temperature range (dT). In the revised version, we have added it in the Methods.

(ii) We have added the zero line in each subfigure in Fig. 2 in this version.

Comment 5: *Can the authors show the microstructure evolution (alpha-Fe precipitation process) during thermal annealing? In addition, if 2:17 type structure has lower energy, why the B-C SRO appear in Fig. 1. What is the relation between B-C SRO and $\text{Er}_2\text{Fe}_{17}$ phase? More importantly, how B modifies the structure of ErFe_{10} to form $\text{Er}_2\text{Fe}_{14}\text{B}$? This should be discussed in more detail.*

Reply: Thanks for your insightful suggestion and comments. (i) In the revised version, we have added the microstructural evolution at different stages (0d, 1d, 3d, 4d, and 5d) in supporting information. As shown in Fig. R16, the precipitation starts from the grain boundary to the interior of the grain. The solid-phase reaction is a relatively slow process and it can be completely precipitated on the fifth day, which explains that the size of the precipitated phase a is not so uniform on the nanometer scale. The precipitated particles may further aggregate and grow.

(ii) The 2:17-type has a lower binding energy than SRO and indeed has a more stable structure. In our previous work, we only got the 2:17 type structure, and no SRO was observed^{4, 5}. In this work, our stoichiometric ratio (1:10) deviates from 2:17 so that anti-site occupation may occur, at the same time, we adopted relatively higher temperature heat treatment conditions (1200°C) to make SRO appear.

Fig. R16 The microstructural evolution in at different stages (0d, 1d, 3d, 4d and 5d)

(iii) The standard 2:17 is a hexagonal structure, in which the Er atom has two crystallographic sites: Er_{2b} and Er_{2d} . The short-range chemical order refers that a single Fe atom replaces the Er_{2b} site hierarchically to form B-C-B-C (C: Er_{2b} replaced by a single Fe atom; B: Er_{2b} not replaced) layers along the c axis (Fig. R16). (iv) As for how the B atoms modulate the structure of $ErFe_{10}$ to $Er_2Fe_{14}B$, we also are very interested, but we have not analyzed the relationship between the two from the perspective of the structure.

Fig. R16 The structure model of the ErFe_{10} phase.

1. Zhou C., *et al.* Near-zero thermal expansion of $\text{ZrW}_2\text{O}_8/\text{Al-Si}$ composites with three dimensional interpenetrating network structure. *Compos. Part B-Eng.* **211**, 108678 (2021).
2. Pang X., Song Y., Shi N., Xu M., Zhou C., Chen J. Design of zero thermal expansion and high thermal conductivity in machinable xLFCs/Cu metal matrix composites. *Compos. Part B-Eng.* **238**, 109883 (2022).
3. Kou X. C., *et al.* Magnetic anisotropy and magnetic phase transitions in R_2Fe_{17} with R= Y, Ce, Pr, Nd, Sm, Gd, Tb, Dy, Ho, Er, Tm and Lu. *J. Magn. Magn. Mater.* **177**, 1002-1007 (1998).
4. Cao Y., *et al.* Ultrawide temperature range super-Invar behavior of $\text{R}_2(\text{Fe,Co})_{17}$ materials (R = Rare Earth). *Phys. Rev. Lett.* **127**, 055501 (2021).
5. Cao Y., *et al.* Manipulating spin alignments of $(\text{Y,Lu})_{1.7}\text{Fe}_{17}$ Intermetallic compounds via unusual thermal pressure. *Inorg. Chem.* **59**, 5247-5251 (2020).

REVIEWERS' COMMENTS

Reviewer #1 (Remarks to the Author):

The revised version can be published as it is.

Reviewer #2 (Remarks to the Author):

The authors addressed most of my questions and I do not have any question now.

Reviewer #3 (Remarks to the Author):

The authors have answered all my questions properly and supplemented several figures. The quality of the manuscript have improved a lot. I have no other questions.